The Author(s) *BMC Pregnancy and Childbirth* 2017, **17**(Suppl 2):362

**RESEARCH**                                                                 **Open Access**

# First birth and the trajectory of women's empowerment in Egypt

Goleen Samari

## Abstract

**Background:** Women's empowerment is often used to explain changes in reproductive behavior, but no consideration is given to how reproductive events can shape women's empowerment over time. Fertility may cause changes in women's empowerment, or they may be mutually influencing. Research on women's empowerment and fertility relies on cross-sectional data from South Asia, which limits the understanding of the direction of association between women's empowerment and fertility in other global contexts. This study uses two waves of a panel survey from a prominent Middle Eastern country, Egypt, to examine the trajectory of women's empowerment and the relationship between first and subsequent births and empowerment over time.

**Methods:** Using longitudinal data from the 2006 and 2012 Egyptian Labor Market Panel Survey, a nationally representative sample of households in Egypt, for 4660 married women 15 to 49 years old, multilevel negative binomial, ordinary least squares, and logistic regression models estimate women's empowerment and consider whether a first and subsequent births are associated with empowerment later in life. Women's empowerment is operationalized through four measures of agency: individual household decision-making, joint household decision-making, mobility, and financial autonomy.

**Results:** A first birth and subsequent births are significantly positively associated with all measures of empowerment except financial autonomy in 2012. Women who have not had a birth make 30% fewer individual household decisions and 14% fewer joint household decisions in 2012 compared to women with a first birth. There is also a positive relationship with mobility, as women with a first birth have more freedom of movement compared to women with no births. Earlier empowerment is also an important predictor of empowerment later in life.

**Conclusions:** Incorporating the influence of life events like first and subsequent births helps account for the possibility that empowerment is dynamic and that life course experiences shape women's empowerment. This and the notion that empowerment builds over time helps portray women's lives more completely, demonstrates the importance of empowerment early in the life course, and addresses issues of temporality in empowerment fertility research.

**Keywords:** Empowerment, Women's agency, Fertility, Women's health, Middle East and North Africa, Egypt

## Background

Fertility in Egypt has been rising since 2008 and is at a two-decade high of 3.5 births per woman [1, 2]. Egypt is also consistently ranked among the worst countries for gender equality — 136 out of 145 counties worldwide in 2015 [3]. Promotion of gender equality and women's empowerment are strategies used to lower fertility [4, 5]. However, the relationship between lower fertility and women's empowerment is reciprocal: greater empowerment

may increase reproductive autonomy, and greater reproductive autonomy may increase women's empowerment. Little consideration is given to how reproductive events can shape empowerment over time.

A recent review of empowerment and fertility found that empowerment is associated with lower fertility [6], greater birth spacing [7], greater contraceptive use [8], lower ideal family size and fertility preferences [9], and increased access to maternal health care [10, 11]. All but one study considered empowerment as the exposure, rather than an outcome of fertility [12]. Fertility changes household dynamics and women's lives, which may

Correspondence: Goleen.Samari@ucsf.edu
Department of Obstetrics, Gynecology & Reproductive Sciences, University of California San Francisco, San Francisco, CA, USA

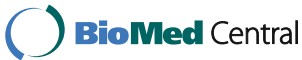

influence women's empowerment. However, the majority of the studies of empowerment and fertility are cross sectional, which limits understanding of the temporality of the relationship, and no studies consider the relationship between fertility and empowerment in the Middle East and North Africa [11]. This study addresses these gaps and takes a longitudinal approach by using two waves of the Egyptian Labor Market Panel Survey (ELMPS), 2006 and 2012, to examine how women's empowerment changes over a 6-year period and, specifically, how women's fertility affects empowerment over time in Egypt.

### Defining empowerment

Women's empowerment is the process in which women acquire enabling resources, like education, which may enhance women's agency, or the ability to define life choices in an evolving historic and social context [13]. Agency includes the ability to formulate one's own strategic choices, to control resources, and to make attitudinal changes under evolving constraints [14]. While there are several related terms for agency, including gender equality and women's autonomy, agency is a context-specific, multidimensional construct, operating at individual and collective levels, and it has been validated in the Egyptian context [14, 15].

This concept of empowerment as an interplay between resources and agency is multidimensional and allows it to be exercised in several life domains, including economic, sociocultural, interpersonal, political, and psychological domains [16–18]. Many studies have demonstrated that women may be empowered in one domain but not in others [19–21]. For example, women can have access to financial resources, but no capacity to make household decisions. As such, this study incorporates several measures of both agency and resources to fully capture the multidimensional aspects of empowerment.

This definition also implies that women's empowerment is a dynamic multilevel process. Important individual life resources and experiences like age, education, and marriage can affect women's empowerment [22, 23]. Family, community, regional, and macro sociopolitical environments are also likely to have a crucial effect on constraining or facilitating women's empowerment [24]. In Nepal and Egypt, women's empowerment varies considerably by region depending on dominant norms and social structure [25, 26]. Empowerment is dynamic and changes over time based on events and experiences at the individual, household, and community levels [26]; therefore, it makes sense that reproductive events, like fertility, would change women's empowerment over time. At later stages in the life course, women's empowerment could be determined by changing circumstances like fertility.

### Fertility and empowerment

Theories of how fertility could affect women's empowerment are built around the notion that reproductive capacity is a central part of women's identities in most societies. It would make sense then that reproductive capacity would shape quality of life, access to resources, and decision-making capacities. Women's sexuality is used to restrict their mobility and access to financial resources [12]. This suggests that women's reproductive capacity, one expression of sexuality, may be linked to their empowerment.

Reproduction encompasses more than the single event of having a child. It triggers a set of other future choices. Reproductive events prompt future decisions about employment, practices surrounding motherhood and childcare, and dynamics within a household. Research shows that more empowered women have fewer children [11], which implies that an underlying assumption may be that women who have higher fertility are less empowered later in life. However, reproductive choice may be a trade-off for other sources of power that bearing a child might give a woman [27]. In some cases, having children can increase women's empowerment by raising their value in society and to their families. Some research shows that as women have sons, they have greater agency within the household [28, 29]. In other cases, motherhood can exacerbate gender inequalities derived from gender roles. In Nepal, if the first born is a boy, then the wife has more influence and relatively fewer children [30]. In India, abortion is associated with less empowerment, and earlier empowerment is a predictor of empowerment later in life [12]. Overall, the relationship between fertility and prediction of women's empowerment has been vastly understudied. The purpose of this study is to address this gap and examine how women's empowerment changes over time and whether women's fertility is associated with greater empowerment over time in Egypt.

### The Egyptian context

Egypt is characterized by rising fertility and limited employment opportunities for women [1, 31, 32]. Egypt is the largest and most densely settled country in the Arab world. Administratively, Egypt is divided into 26 governorates grouped together as the Urban Governorates (Cairo, Alexandria, Port Said, Suez), the governorates of Upper and Lower Egypt (Lower Egypt is primarily the Nile delta, while Upper Egypt is south of the Nile delta), and the Rural Governorates (including all other parts of the country). There are marked regional heterogeneities. About 25% of the Egyptian population lives in Upper Egypt, which includes more than 90% of the poorest villages, characterized by higher fertility and unmet need for contraception [33, 34].

The Social Institutions and Gender Index, which measures legislation, practices, and attitudes that restrict women's rights, classified Egypt among the countries that are "very high" in gender discrimination in 2014 [35]. In 1981, Egypt signed the Convention on the Elimination of All Forms of Discrimination Against Women (CEDAW), which guarantees women equal rights across different dimensions [36]. Despite modernization, the traditional role of women in Egyptian society has not changed [37]. Only 22% of women participate in the labor force, and the family and household context remains the main source of financial support for most Egyptian women [36, 38].

Marriage for women remains universal, with fewer than 2% having never married by age 40 [39]. Women are typically married between 18 and 23 years old [39]. Marriages are typically arranged by parents, with the interests not only of the young adults but also the family at large in mind [22]. In Egypt, dowries allow families to attract husbands of equal social standing [40]. A dowry can increase the household economic resources as well as the wife's participation in the expenditures and other household decisions [41].

Child marriage remains prevalent, especially among the poorest 20%, in which more than a third of women are married before age 18 [22]. Women who marry at a young age generally have fewer personal financial resources than their older counterparts, tend to be more dependent on their husbands, and have a lower social standing in the household [42, 43]. However, age at marriage should not be a proxy for women's post-marital agency in Egypt [22]. Endogamy is also common in Egyptian society, as women are often married to cousins or other close relatives. Close premarital family ties between natal and marital families may increase a woman's agency within her marital family [44]. Newly married couples often live with the husband's family, and intergenerational co-residence in extended family households is common [45].

Women take on traditional roles in the household and spend the majority of their time physically inside of their homes; outside labor force participation is infrequent [39]. Families are organized along patriarchal lines: married men are heads of households and make decisions for the household and its members [45]. However, this attitude is changing: young adults in some parts of Egypt say that women and men should share household decisions [46]. Control over household decisions, or lack thereof, is an important and direct measure of women's agency within Egyptian families [14]. Women report very little agency, as fewer than 15% of women make household decisions about health care, large purchases, or visits to family and friends [31, 47]. Similar to other settings, more educated women in Egypt have more agency

[36]. However, empowering women will involve more than simply providing greater education and employment opportunities and, instead, require changing community gender norms and values [26].

Women's value is strongly tied to fertility, and norms for women include premarital virginity and marital fertility [48, 49]. In Egypt, fertility had declined from 7.2 births per woman in the early 1960s to 5.3 births by 1980 to 2.8 births by 2008 [9]. However, by 2015 fertility had risen to 3.5 births per woman, indicating that Egypt's demographic transition has stalled [1, 2]. Region of residence and urban residence are important determinants of fertility in Egypt, as fertility is higher in rural Upper Egypt [37, 50]. In 2005, 23% of 19-year-old women were already mothers or pregnant [48]. Egypt is characterized by first births soon after marriage and aversions to one-child families [51]. There is also a gender preference for sons [52]. Fewer than 1% of births are outside of marriage.

Very few studies consider the relationship between women's agency and reproductive health in Egypt. Studies find that greater participation in household decisions and greater freedom of movement are associated with use of modern contraception and better health of infants yet higher fertility [53–57]. Studies on other women's health outcomes find that a lack of agency is associated with greater anxiety [31, 58] and more exposure to intimate partner violence [59]. The paucity of empowerment in Egypt is also associated with lower employment rates [36]. However, little is known about the relationship between empowerment and fertility in Egypt, specifically, how fertility may shape women's agency over time. When women's social roles are strained, reproductive events might provide household power and are important to consider regarding women's empowerment in Egypt.

### Research aims

The focus is on two research aims: (1) Are first and subsequent births associated with greater empowerment over time? (2) How does women's empowerment change over time? Given the value of women tied to fertility in Egypt, I hypothesize that women who have a birth will have greater empowerment across several different dimensions. I also expect to find that women gain empowerment over time.

### Methods
### Data
The ELMPS is a nationally representative panel survey of households in Egypt undertaken by the Central Agency for Public Mobilization and Statistics and the Economic Research Forum. Data were collected at three points in time: Wave 1 data were collected in 1998,

Wave 2 data were collected in 2006, and Wave 3 data were collected in 2012. Wave 1 data (1998) do not include women's fertility and empowerment. Waves 2 and 3 (2006 and 2012) of ELMPS include data on fertility and empowerment for a large nationally representative sample of married women. Of all the 37,140 individuals interviewed in 2006, 28,770 (77%) were successfully re-interviewed in 2012. Attrition was primarily from the Cairo region, and an analysis of panel attrition has been discussed elsewhere [32].

All data were self-reported during a face-to-face interview conducted by a trained field interviewer [32]. The ELMPS data contain individual-level information about education, age, gender, and many other demographic variables as well as household-level information about assets and consumption and location. The ELMPS measures of empowerment include (1) questions on participation in household decision-making, (2) questions about a woman's ability to move around on her own (mobility), and (3) access to financial resources.

The analytic sample is restricted to married women in their childbearing years in 2006 with complete data on fertility and empowerment in the 2006 and 2012 ELMPS. In the 2006 ELMPS, 49% or 18,555 are women, 26.8% or 9937 are in their childbearing years (between the ages of 15 and 49), and 6296 women are married. Of the married women, 108 are missing data on empowerment and fertility in 2006 and 455 women are missing data on their spouses. Between 2006 and 2012, 1181 women were lost to follow-up, and compared to the analytic sample and consistent with an analysis of attrition, these women are slightly younger, higher educated women from the Greater Cairo area (Additional file 1: Table S1) [32]. The final analytic sample includes 4660 married women in their childbearing years.

## Measures
### Dependent: empowerment
Four measures of empowerment: individual household decision-making, joint household decision-making, mobility, and financial autonomy capture women's empowerment in 2012. Decision-making and mobility are measures of agency, and financial autonomy is a measure of resources. For *household decision-making*, respondents were asked who in the family had final say on a series of decisions including (1) making large household purchases, (2) making household purchases for daily needs, (3) visits to family, friends, or relatives, (4) food that should be cooked each day, (5) getting medical treatment or advice for the woman herself, (6) buying clothes for herself, (7) taking a child to the doctor, (8) sending children to school, (9) sending children to school on a daily basis, (10) buying clothes for children. The reliability coefficient is 0.74, implying a

reasonable to high level of correlation among these items. Response categories include the respondent alone, husband, respondent and husband jointly, in-laws, respondent, husband, and in-laws jointly, or others. Items are recoded so that 6 = respondent, 5 = jointly by respondent and husband, 4 = jointly by respondent, husband, and in-laws, 3 = husband, 2 = in-laws, and 1 = others. Since these response categories do not create an interval, two count variables capture household decision-making: a count of the number of times the respondent herself makes decisions, *individual household decision-making*, and a count of the number of times the respondent and somebody else within the household participate in decisions, *joint household decision-making*. These two measures capture the different ways a respondent has a say in household decisions without giving more preference for a decision made alone and a decision made with a partner or other family member. When more value is given to women making decisions alone, there is an assumption that more agency means making a decision on your own. However, greater agency may include the ability to discuss and negotiate decisions with a partner or family member. The two outcome measures are zero sum: if women do not make *individual* or *joint household decisions*, they make no household decisions. For both variables, counts range from 0 to 10, with a higher count indicating more participation on a greater number of household decisions.

*Mobility* is defined as a continuous measure, based on four items in the ELMPS assessing the respondents' ability to leave the house. For mobility, respondents were asked whether they could go to a local market, health center, or home of relatives or friends in the neighborhood, and if they could take children to a health center. Responses were reverse coded and included 4 = without permission, 3 = just inform them, 2 = need permission, and 1 = cannot go alone, so that higher scores indicated greater control in personal mobility decisions. Items were averaged to create a scale from 1 to 4, with higher responses indicating a higher amount of personal control in mobility decisions ($\alpha$ = .79).

*Financial autonomy* is a dichotomous variable based on two items. Respondents were asked Do you have direct access to household money in your hand to use? and Do you personally have savings, own land, a house, jewelry, or other valuables which you can sell or use as you please? The responses to these two questions were combined: those who responded "yes" to one or both items are defined as having access to financial resources, while those who responded "no" on both do not.

## Fertility
*Number of births* is a count of all live births reported in the woman's birth history. To test the difference between

those who have a first birth and those who have no births, *number of births* is coded as a categorical variable with $0 = 0$ births, $1 = 1$ birth, $2 = 2$ births, $3 = 3$ births, and $4 = 4$ or more births. In multivariate models, one birth is used as the reference category so as not to use the smallest sample category of zero births and to show whether there is something unique about first births compared to multiple births.

### Covariates

Sociodemographic covariates that could be endogenous to fertility and empowerment are accounted for to ensure the sociodemographic variables are not confounding the explanations for primary hypotheses. Age, age at marriage, value of dowry, relationship to the husband, education, having ever worked, region, household wealth, and spouse's age and education were added to the model to check for spuriousness or redundancy. Woman's age is measured as age in years at the time of the interview. Education is defined as years of completed education. Age at marriage is the age at which the woman was married and dichotomized as less than 18 years old and 18 years or older at age of marriage. The value of the dowry is a categorical measure indicating whether the respondent had no dowry or some dowry. I also include a category reflecting non-response for this variable, because a sizable proportion of women did not give an answer. A woman's relationship to her husband is a categorical variable that captures whether the respondent is related to her husband or not. Having ever worked is a dichotomous variable indicating whether or not a woman has ever worked for pay.

Region is coded $0 =$ Greater Cairo, $1 =$ Alexandria and Suez, $2 =$ Urban Lower Egypt, $3 =$ Rural Lower Egypt, $4 =$ Urban Upper Egypt, and $5 =$ Rural Upper Egypt. The household wealth index is estimated from asset variables using principal component analysis. Ownership of consumer items such as a TV or car as well as characteristics of the dwelling such as flooring and roofing and types of access to water and sanitation are used. Household wealth scores are divided into quintiles: poorest, poor, middle, rich, and richest.

Communities are operationalized as the ELMPS primary sampling unit (PSU). For the ELMPS, all villages in rural areas or urban quarters (shiyakhas) in cities were listed and assigned weights based on their population. The selected shiyakhas and villages are divided into primary sampling units of 1500 housing units in each, and then one or more PSUs are selected from each village or shiyakha. Each of the 26 governorates in Egypt is allocated a number of PSUs in the master sample that is proportionate to its size and its urban/rural distribution. There are 418 PSUs with an average of 11 observations per cluster.

### Analytic strategy

The analysis has two parts. First, univariate and bivariate associations between all variables were estimated and revealed no concerns for collinearity among the covariates. Preliminary bivariate results show that there are no significant differences in any measure of empowerment by gender of the first birth. For parsimony, these analyses and results are not included. Average empowerment by year and by number of births is calculated and presented.

In the second part of the analysis, I estimated a series of multilevel models which control for the correlation among women resulting from clustering within PSU and enable tests for differences in the community effects for women. Four separate models are estimated, one for each of the dependent variables: individual household decision-making, joint household decision-making, mobility, and financial autonomy. In each model, the first level is the individual, and the second level is the primary sampling unit (as a proxy for community). Multilevel mixed effect negative binomial regression models are used for decision-making. Due to over-dispersion in the decision-making outcomes, tests of model fit favored negative binomial regression models, which allow for the variance to be greater than the mean. Zero-inflated Poisson models and negative binomial regression models of a combined count measure of both individual and joint household decisions were also checked and produced similar results (Additional file 1: Table S2). Mixed effect ordinary least squares (OLS) models are used for mobility, and logistic mixed effect multilevel models for financial autonomy. All models of empowerment in 2012 also account for empowerment in 2006. For a sensitivity analysis, multilevel mixed effects models predicting each empowerment outcome defined as changes in empowerment between 2006 and 2012 (i.e., the difference in number of individual household decisions made in 2006 and 2012) were also examined. All models were estimated in STATA 14.

### Results

Table 1 shows the fertility and demographic characteristics of married women in Egypt. On average, women have 2.79 births. Only 10% of women have not had a birth, and 30% of women have had four or more live births. About half of first births are girls. While the average number of births is close to three, the women have anywhere from 0 to 13 births. The women are between 15 and 49 years old, and the average age is 32 years. On average, women have between 7 and 8 years of education. The average age at marriage is 20 years. A third of women report having ever been employed, and a third are related to their husbands, most often as first cousins. Half of the women live in urban areas with 10% in the Greater Cairo region.

**Table 1** Fertility and sample characteristics (means (SD) or %) of married women ages 15–49, Egyptian Labor Market Panel Survey

| Key variables in 2006 | Married women, N = 4660 | |
|---|---|---|
| | Number (N) | % or mean (SD) |
| Number of births | | |
| 0 | 458 | 9.83 |
| 1 | 827 | 17.8 |
| 2 | 1073 | 23.0 |
| 3 | 909 | 19.5 |
| 4+ | 1393 | 29.9 |
| Mean (standard deviation, SD) | 4660 | 2.79 (1.99) |
| Gender of first birth | | |
| Boy | 1993 | 47.5 |
| Girl | 2203 | 52.5 |
| Current age (years) | 4660 | 32.2 (8.70) |
| 15–24 years | 1086 | 23.3 |
| 25–34 years | 1670 | 35.8 |
| 35–44 years | 1392 | 29.9 |
| 45–49 years | 512 | 11.0 |
| Years of education | 4660 | 7.29 (5.69) |
| Age at marriage (years) | 4660 | 20.4 (4.02) |
| Less than 18 years old | 1113 | 23.9 |
| 18 years or older | 3547 | 76.1 |
| Value of dowry | | |
| No response | 1251 | 26.9 |
| No amount | 1798 | 38.6 |
| Some amount | 1611 | 34.6 |
| Related to husband (1/0) | 1488 | 32.0 |
| Ever worked (1/0) | 1416 | 30.4 |
| Region | | |
| Greater Cairo | 470 | 10.1 |
| Alexandria and Suez Canal | 394 | 8.45 |
| Urban Lower | 576 | 12.4 |
| Urban Upper | 734 | 15.8 |
| Rural Lower | 1388 | 29.8 |
| Rural Upper | 1098 | 23.6 |
| Household wealth index | | |
| Poorest | 879 | 18.9 |
| Poorer | 1024 | 22.0 |
| Middle | 1030 | 22.1 |
| Richer | 892 | 19.1 |
| Richest | 835 | 17.9 |
| Husband's age in years | 4660 | 39.1 (9.74) |
| Husband's years of education | 4660 | 8.68 (5.47) |

Women's spouses have on average a year more education than the women and are on average 39 years of age.

Table 2 summarizes the outcome variables and shows the average empowerment for married women in 2006 and 2012. The results are similar in both years, with some exceptions. In general, respondents have a low amount of personal control in household decisions. The average score for respondents participating in household decisions in 2006 is equivalent to participating in only two to three decisions out of a total of 10 (mean = 2.48, standard deviation (SD) = 2.25). Nonetheless, there is still variation, with scores covering the full range from 0 to 10. While individual decision-making is statistically significantly different in 2012, the average score is still equivalent to participating in two to three decisions out of 10 (mean = 2.54, SD = 2.76). For respondents making household decisions along with someone else, the average score is slightly higher and equivalent to jointly making three to four decisions. Joint decision-making significantly decreases over time to close to three decisions in 2012 ($p < 0.001$). Twenty-one percent ($N = 989$) of women make no household decisions (no individual or joint decisions) in the 2012 survey. On the other hand, average mobility significantly increases over time ($p < 0.001$), although the average score is equivalent to a response between "need permission" and "just inform them," indicating that most women need permission to go outside of the home. Another change over the 6-year period is in access to financial resources: in 2006, 65% of women report having access, while in the 2012 survey, fewer women or 60% of women report having access ($p < 0.001$).

Table 3 shows the average empowerment in 2006 and 2012 by number of births in 2006. All measures of empowerment in 2006 and 2012 significantly differ by number of births. The means and frequencies for each empowerment measure at each number of births are significantly different between 2006 and 2012 ($p < 0.001$). In 2012, the average number of individual household decisions and mobility increase for each birth from zero

**Table 2** Mean (SD) of empowerment for married women, 2006 and 2012 Egyptian Labor Market Panel Survey, N = 4660

| Key empowerment measures | Married women | | |
|---|---|---|---|
| | Range | 2006 | 2012 |
| Household decision-making | | | |
| Individual decision-making*** | 0–10 | 2.48 (2.25) | 2.54 (2.76) |
| Joint decision-making*** | 0–10 | 3.57 (2.64) | 3.05 (3.04) |
| Mobility*** | 0–4 | 2.05 (0.68) | 2.55 (0.81) |
| Financial autonomy*** | 0–1 | 0.65 (0.48) | 0.60 (0.49) |

*$p < 0.05$, **$p < 0.01$, ***$p < 0.001$ for significant differences between 2006 and 2012

**Table 3** Means (SD) or % of married women's empowerment in 2006 and 2012 by births in 2006, Egyptian Labor Market Panel Survey (N = 4660)

| Number of births 2006 | | 2006 individual decision-making*** | 2006 joint decision-making*** | 2006 mobility*** | 2006 financial autonomy** |
|---|---|---|---|---|---|
| | | Mean (SD) | | | % |
| 0 | 458 | 1.31 (1.38) | 2.58 (1.91) | 1.48 (0.56) | 57.6 |
| 1 | 827 | 1.96 (1.96) | 3.91 (2.53) | 2.02 (0.66) | 61.2 |
| 2 | 1073 | 2.62 (2.18) | 3.88 (2.61) | 2.14 (0.65) | 66.2 |
| 3 | 909 | 2.92 (2.40) | 3.76 (2.79) | 2.14 (0.66) | 66.6 |
| 4+ | 1393 | 2.81 (2.43) | 3.29 (2.73) | 2.12 (0.68) | 67.3 |
| | | | | | |
| Number of births 2006 | | 2012 individual decision-making*** | 2012 joint decision-making*** | 2012 mobility* | 2012 financial autonomy*** |
| 0 | 458 | 1.80 (2.07) | 3.35 (2.82) | 2.34 (0.81) | 51.7 |
| 1 | 827 | 2.93 (2.68) | 3.78 (3.02) | 2.54 (0.79) | 56.6 |
| 2 | 1073 | 3.05 (2.87) | 3.49 (3.05) | 2.62 (0.81) | 63.8 |
| 3 | 909 | 2.62 (2.83) | 3.05 (3.05) | 2.58 (0.81) | 63.8 |
| 4+ | 1393 | 2.07 (2.76) | 2.20 (2.95) | 2.57 (0.82) | 60.5 |

$*p < 0.05$, $**p < 0.01$, $***p < 0.001$ for differences in empowerment measures by number of births
All means and percentages also significantly differ between 2006 and 2012 at $p < 0.001$

to two and then decline for three and four or more births. Financial autonomy in 2012 has a similar relationship with number of births: peaking at two births, remaining stable for those who had three births, and declining for four or more births. In 2012, joint household decision-making peaks for those who had one birth and then declines.

Table 4 shows the multilevel negative binomial, OLS, and logistic models of empowerment in 2012, net of any controls. For the negative binomial, incident rate ratios are presented and can be interpreted as a percent change in the incident rate. For the logit models, odds ratios are presented for each explanatory variable. For all outcomes, the likelihood-ratio test comparing the multilevel models with a standard regression model confirms that a multilevel model is preferred.

Women who have not had a birth make 26% fewer individual household decisions in the 2012 survey compared to women with a first birth, all else constant ($p < 0.001$). A first birth is significantly associated with all measures of empowerment except financial autonomy in 2012. However, in the model of financial autonomy, women's employment is associated with women's financial autonomy in 2012 and may capture some of the effects of parity on financial autonomy. Women who have not had a birth by 2006 also make 10% fewer joint household decisions compared to women with a first birth ($p < 0.05$). There is also a positive relationship with mobility, as women with a first birth have more freedom of movement compared to women with no births ($p < 0.001$).

Each subsequent birth is also associated with greater empowerment in terms of decision-making and mobility

as compared to women with only a first birth ($p < 0.001$). Women who have had four or more births in 2006 make 64% more individual household decisions, 48% more joint household decisions, and have greater mobility compared to women with a first birth, all else constant ($p < 0.001$). There are also significant differences in decision-making and mobility for women who have had two vs. three births and three vs. four births ($p < 0.001$) (not shown). Preliminary analyses with number of births as a continuous measure are consistent with the results presented — for each additional birth reported in 2006, women make more individual and joint household decisions and have greater freedom of movement ($p < 0.001$). Sensitivity analyses of an alternative outcome, changes in women's empowerment between 2006 and 2012, are also consistent with the results presented.

Women who make more individual decisions in 2006 also have greater mobility in 2012 and 6% higher odds of being financially autonomous in 2012. Women who make more joint household decisions in 2006 make more individual decisions and have more financial autonomy in 2012 ($p < 0.001$). Women with more mobility in 2006 make more individual household decisions and have more mobility and financial autonomy in 2012 ($p < 0.001$). Financial autonomy in 2006 is associated with individual decision-making and financial autonomy in 2012 ($p < 0.01$). No empowerment measure in 2006 predicts joint household decision-making in 2012.

Women's empowerment varies by individual characteristics, communities, and regions. Younger women are associated with greater empowerment over time, because older women make fewer individual and joint household decisions and have less mobility. Women with more

**Table 4** Negative binomial, OLS, and logistic multilevel regression models of married women's empowerment in 2012; 2006 and 2012 Egyptian Labor Market Panel Survey (N = 4660)

| Key variables | 2012 individual decision-making Negative binomial | | 2012 joint decision-making | | 2012 mobility OLS | | 2012 financial autonomy Logistic | |
|---|---|---|---|---|---|---|---|---|
| | IRR | (SE) | IRR | (SE) | β | (SE) | OR | (SE) |
| Births 2006 (Ref = 1 birth) | | | | | | | | |
| 0 | 0.74*** | (0.05) | 0.90* | (0.06) | −0.13** | (0.05) | 1.12 | (0.15) |
| 2 | 1.30*** | (0.07) | 1.23*** | (0.07) | 0.08* | (0.04) | 1.20 | (0.13) |
| 3 | 1.49*** | (0.09) | 1.41*** | (0.09) | 0.09* | (0.04) | 1.18 | (0.15) |
| 4+ | 1.64*** | (0.12) | 1.48*** | (0.10) | 0.18*** | (0.05) | 1.23 | (0.18) |
| Respondent household decision-making 2006 | 1.06*** | (0.01) | 1.01 | (0.01) | 0.03*** | (0.01) | 1.06** | (0.02) |
| Joint household decision-making 2006 | 1.04*** | (0.01) | 1.01 | (0.01) | 0.01 | (0.01) | 1.06*** | (0.02) |
| Mobility 2006 | 1.19*** | (0.04) | 1.00 | (0.03) | 0.10*** | (0.02) | 1.33*** | (0.09) |
| Financial autonomy 2006 | 1.10* | (0.04) | 1.02 | (0.04) | −0.03 | (0.03) | 1.18* | (0.09) |
| Age (years) | 0.93*** | (0.00) | 0.94*** | (0.00) | −0.02*** | (0.00) | 1.00 | (0.01) |
| Years of education | 1.02*** | (0.00) | 1.02*** | (0.00) | 0.01* | (0.00) | 1.03** | (0.01) |
| Less than 18 years old at marriage | 0.87** | (0.04) | 0.85*** | (0.04) | −0.10** | (0.03) | 0.94 | (0.09) |
| Dowry (Ref = none) | | | | | | | | |
| No response | 0.92 | (0.05) | 0.99 | (0.05) | 0.04 | (0.04) | 1.01 | (0.11) |
| Some | 0.92 | (0.04) | 1.05 | (0.05) | −0.01 | (0.03) | 1.10 | (0.10) |
| Related to husband | 1.00 | (0.04) | 0.98 | (0.04) | −0.04 | (0.03) | 0.91 | (0.07) |
| Ever worked | 1.08 | (0.04) | 1.09* | (0.04) | 0.11*** | (0.03) | 1.59*** | (0.14) |
| Region (Ref = Greater Cairo) | | | | | | | | |
| Alexandria and Suez Canal | 0.90 | (0.09) | 1.06 | (0.10) | −0.31*** | (0.08) | 0.70 | (0.16) |
| Urban Lower | 0.79* | (0.07) | 1.17 | (0.10) | −0.09 | (0.07) | 0.69 | (0.15) |
| Urban Upper | 0.61*** | (0.06) | 1.16 | (0.10) | −0.22** | (0.07) | 0.29*** | (0.06) |
| Rural Lower | 0.75*** | (0.06) | 1.22* | (0.09) | −0.06 | (0.06) | 0.49*** | (0.09) |
| Rural Upper | 0.62*** | (0.06) | 0.94 | (0.08) | −0.25*** | (0.07) | 0.33*** | (0.07) |
| Household Wealth Index (Ref = poorest) | | | | | | | | |
| Poorer | 0.91 | (0.05) | 0.96 | (0.05) | −0.11** | (0.04) | 0.88 | (0.10) |
| Middle | 0.91 | (0.06) | 0.94 | (0.06) | −0.06 | (0.04) | 0.90 | (0.11) |
| Richer | 0.87* | (0.06) | 1.00 | (0.07) | −0.13** | (0.05) | 0.68** | (0.09) |
| Richest | 0.85* | (0.07) | 0.93 | (0.07) | −0.13* | (0.05) | 0.71* | (0.11) |
| Husband's age (years) | 1.01* | (0.00) | 0.99* | (0.00) | 0.00 | (0.00) | 1.01 | (0.01) |
| Husband's years of education | 0.99* | (0.00) | 1.01* | (0.00) | −0.01* | (0.00) | 0.99 | (0.01) |
| Variance at level 1 (individual level) | | | | | 0.57 | (0.01) | | |
| Variance at level 2 (PSU level) | 0.42 | (0.08) | 0.55 | (0.09) | 0.06 | (0.02) | 0.02 | (0.01) |

*p < 0.05, **p < 0.01, ***p < 0.001
Standard errors in parentheses
*OLS* ordinary least squares, *IRR* incidence rate ratio, *SE* standard error, *PSU* primary sampling unit

educational attainment and those who have worked are more empowered compared to those who have not worked. Women who married before the age of 18 make fewer individual and joint household decisions and have less mobility compared to those who married at 18 years or older. There are no significant differences in empowerment by dowry or relation to husbands. Variance at the community level explained some of the relationship between fertility and empowerment over time. Additionally, there are also important regional differences. Women who live in both rural and urban Upper Egypt are less empowered compared to women who live in Greater Cairo. Women in rural Lower Egypt make fewer individual household decisions, more joint household decisions, and have 51% less financial autonomy compared to women in Greater Cairo.

## Discussion

This is the first study to explore whether fertility is associated with empowerment over time, and how women's empowerment changes with important life events like a first birth in an important Middle Eastern context. The focus on fertility in Egypt is salient, given its rising fertility since 2008 [1, 2]. Promotion of gender equality is a long-standing goal of the international development organizations (e.g., the World Bank), because it is positively associated with lower fertility and better health for women and children [60]. Women's empowerment is a known determinant of lower fertility, but little is known about how fertility affects empowerment [11]. This study extends work on women's empowerment and health by examining how women's empowerment changes over time and whether first and subsequent births affect diverse measures of women's empowerment in Egypt.

As hypothesized, women who have had a birth in 2006 are more empowered in 2012. Women who have had a first and subsequent births by 2006 make more individual and joint household decisions and have greater mobility. The positive association aligns with work that shows having a son leads to greater household agency [28, 29]. However, there were no differences in empowerment by gender of the children. These results are in contrast to those of the only other study that considers reproductive events and empowerment, which found no associations between unwanted and mistimed pregnancies and empowerment [12]. The multivariate results also slightly vary from the bivariate findings that indicate women make gains in agency for each birth until two or three births and then have a decline in household agency for subsequent births. In Egypt, the social expectation of women is to have three births [9], so it would follow that women's gains in empowerment from number of births would peak at close to three births as they meet that expectation, and then they make fewer or no gains in empowerment for each subsequent birth. Nonetheless, when adjusted for covariates, which capture multiple aspects of women's lives, first and all subsequent births are associated with more empowerment over time. It is not surprising that Egyptian women experience greater empowerment after bearing a child, because their value is tied to their fertility [49]. In most countries, bearing children is associated with more bargaining power for married women [17]. Given the strong implications of women's public behavior for family reputation and honor in the Egyptian context and the many norms governing women's sexual behavior, reproductive events are likely to be a source of power.

The birth of a child can be a source of household agency, which shows that within the definition of empowerment, children should also be considered a resource from which agency is derived [13]. Childbearing sets up choices and practices surrounding childcare, which may provide women with more opportunity to foster household control. Importantly, Egypt between 2006 and 2012 was experiencing an economic downturn, and there were even more constrained economic opportunities for women and families [61]. The downward shift in women's financial autonomy between 2006 and 2012 may be an indicator of this economic change. Children can provide security and a sense of purpose in the household for women in the midst of the instability. In settings like Egypt, where women have limited access to other channels of security, children are of greater value for their mothers' current and future security [9]. Furthermore, under various economic conditions, having children, sons in particular, is associated with more bargaining power for married women [17, 28, 29]. This implies that the relationship between fertility and empowerment would likely remain the same in periods of economic expansion, but future research is needed to confirm the association under other economic conditions. These findings suggest that fertility is important for women's position within the family and community in Egypt, and fertility should be included in the study of empowerment over time.

The second hypothesis, that women will gain empowerment over the life course and will be more empowered later in life as compared to earlier in life, is also supported. As expected, earlier empowerment is a predictor of later empowerment. However, not every indicator of empowerment is associated with every measure of empowerment over time. This finding provides further support that each measure of empowerment is demonstrating something different about women's resources and agency. For example, greater joint decision-making in 2006 is associated with more empowerment in 2012 except in terms of mobility. The women who participate in more joint decisions in 2006 could be more communicative with other household members, allowing them to gain resources, like financial autonomy, over time. More mobility in 2006 is associated with all measures of empowerment except joint decision-making in 2012. Perhaps women who are able to leave the house without permission can obtain the necessary resources to exercise control within the household across other dimensions of empowerment. Importantly, even when accounting for earlier empowerment, women's fertility is consistently an important predictor of later empowerment.

The results also show that the relationship between individual fertility and agency is partially explained by regional differences and social conditions at the community level. Women in Upper Egypt are less empowered over time compared to women in other regions. This is consistent with cross-sectional research that finds Upper Egypt is characterized by few economic opportunities, large gender gaps in education, and fewer empowered

women [31, 47, 56]. By demonstrating the significance of regional variation, this study also builds on empowerment research that exclusively focuses on the relationship between women's individual characteristics and their own empowerment [62] and adds to a growing body of literature on the importance of regional variations in both fertility [4, 63, 64] and empowerment [25, 26, 53]. This study shows that even when accounting for fertility, there is regional variation in agency over time and suggests that societal factors in Upper Egypt are not conducive to empowering women over time.

The importance of community-level variation in women's empowerment over time is also consistent with cross-sectional research from Egypt that shows community social norms are an important predictor of women's empowerment at one time point [26]. Oppressive contexts often truncate the range of options that women consider to be viable, thus interfering with their agency and ability to make choices. The measure of community captures women's geographic location, and potentially, the social context of that community. Theoretically, it would be expected that people from the same area might be more similar to each other in terms of attitudes and norms than people from other areas [65]. For example, in five separate Asian countries, the gender context of communities was found to influence the relative weight given to the husbands' and wives' decision-making preferences for contraceptive use [63]. Wider social norms about women's capabilities, duties, and place in society affect household power dynamics and expectations of women in families [24].

In Egypt, women's public behavior has strong implications for family reputation and honor; therefore, empowerment is influenced by fertility and community-specific norms. Norms for women in Egypt include marital fertility and gendered roles tied to childbearing and childrearing [48], which can drive women's fertility and empowerment over time. While results establish that the community is explaining some aspect of the relationship between women's fertility and empowerment in Egypt, the actual pathways for community variation in the relationship between fertility and empowerment are unclear. To clarify this relationship further, other community-level attributes that might affect women's empowerment, like gender norms, district-level rural development expenditure, community programs for women, etc., are needed. Future research should consider what community factors are associated with fertility and empowerment over time. These findings align with work that suggests strategies to enhance women's empowerment need to operate at the community level and expand beyond education, employment, and delayed marriage [64, 66, 67]. Empowering women in Egypt will involve more than simply providing greater education and employment opportunities and,

instead, require changing community norms and values about fertility and gender relations [26].

From a methodological standpoint, the findings highlight the problematic aspects of conducting cross-sectional analyses of empowerment and fertility, as the relationship functions in both directions. Empowerment and fertility research is largely cross sectional [42, 68–70]. This study shows that temporality is an important issue for empowerment fertility research. Even when the measurement of fertility precedes an outcome (e.g., empowerment), causality cannot be established in the absence of controls for external, concurrent secular changes that are also likely to influence these relationships. Longitudinal designs do, however, better reflect the process of women's empowerment and help determine the mechanisms that may facilitate or hinder women's empowerment.

Some limitations of this work are notable. First, the ELMPS lacks empowerment measures from more than two waves of data, which limits the full understanding of lifetime influences of empowerment. The 6-year lag between waves, while well suited to detect stable linkages, may blind the analysis to important temporal inflections in effects. The analysis controls for empowerment in 2006 in the relationship between fertility in 2006 and empowerment in 2012, but the relationship could still contain some reciprocity. That is, childbearing could be a function of being more empowered prior to 2006. Only baseline connections could be established. While the study accounts for temporal ordering, conclusions about causal relationships between fertility and empowerment cannot be made.

Second, while the ELMPS has extensive data on work and fertility history, the data do not include information on women's contraceptive behavior, fertility intentions, or sexual behavior. Measures of women's agency that are more proximate to fertility like contraceptive preferences and decision-making and sexual agency, like the ability to refuse sex, would help clarify the pathway that links fertility to empowerment. Furthermore, existing measures of women's empowerment provide little contextual information on household dynamics. For example, while separating household decision-making into individual and joint decisions contextualizes who is making the decisions, the questions on household decision-making provide little insight into discussions women may have had with partners about those household decisions. Finally, while the observed relationships are quite strong and the models adjusted for a range of factors, the analysis does not account for unobservable factors, which may affect the relationship between fertility and women's empowerment over time. Moreover, linkages may very well vary across (be moderated or mediated by) categories of indicators that were controlled for, like region, education, and employment. Future research should consider

the relationship between fertility and empowerment and intervening variables over several waves of data.

Several strengths of the study outweigh these limitations. This study considered multiple dimensions of empowerment by using several empowerment outcomes. The analysis uses two waves of data from a prospective, panel survey to address weaknesses in cross-sectional studies. By incorporating measures of empowerment at two points in time, the findings provide new evidence of lasting associations between earlier and later empowerment and between fertility and empowerment over time. These associations are examined using a large sample of married women from an important Middle Eastern context and highly contextualized covariates, providing robust and relevant estimates. The findings are generalizable to Egypt and may be relevant for other country contexts with similar fertility rates and gaps in gender equity. The findings also provide support for the use of multilevel models in future fertility and reproductive health research [71]. These features strengthen the interpretations of the findings, enhancing their policy relevance.

## Conclusions

This study uses a panel from two waves of data to better reflect the process of women's empowerment over time. By examining the trajectory of women's empowerment in Egypt, this work shows that women are more empowered over time. This contributes to our understanding of how early empowerment conditions influence later empowerment, and this could imply that interventions should empower girls early to have an effect on their empowerment later in life. The finding that fertility is associated with more empowerment — both individual and joint household decision-making and mobility — shows how women's value being tied to fertility in Egypt has implications for women's power and control in their daily lives. In the most densely populated country in the Middle East, a better understanding of how fertility promotes gains in women's empowerment over time can inform strategies to address the costly burden of high fertility while enhancing women's empowerment.

### Open peer review

Peer review reports for this article are available in Additional file 2.

### Additional files

**Additional file 1: Tables S1 and S2.** (DOCX 15 kb)

**Additional file 2:** Open peer review. (PDF 224 kb)

### Abbreviations

ELMPS: Egyptian Labor Market Panel Survey; PSU: Primary sampling unit

### Acknowledgements

I thank Anne Pebley for helpful comments on a previous version of this paper, and Steven Wallace, Linda Bourque, Judith Seltzer, and Megan Sweeney for general guidance. This research uses data from the Egyptian Labor Market Panel Survey, a program project directed by Ragui Assaad at the University of Minnesota and the Economic Research Forum.

### Funding

This article is part of a special issue on women's health and empowerment, led and sponsored by the University of California Global Health Institute, Center of Expertise on Women's Health, Gender, and Empowerment. It also received feedback at a workshop partially funded by the National Institutes of Health (NIH) National Center for Advancing Translational Sciences (NCATS) University of California, Los Angeles (UCLA) Clinical and Translational Science Institute (CTSI) grant number UL1TR000124. The research was supported by Eunice Kennedy Shriver National Institute of Child Health and Human Development (NICHD) training grants at UCLA (T32HD007545) and the University of Texas at Austin (T32HD007081), and the California Center for Population Research at UCLA (P2C-HD041022) and the Population Research Center at the University of Texas at Austin (R24HD042849), which both receive core support from the NICHD. The content is solely the responsibility of the author and does not necessarily represent the official views of the NIH.

### Availability of data and materials

The datasets analyzed during the current study are available through the Economic Research Forum, http://www.erfdataportal.com/index.php/catalog.

### About this supplement

This article has been published as part of BMC Pregnancy and Childbirth Volume 17 Supplement 2, 2017: Special issue on women's health, gender and empowerment. The full contents of the supplement are available online at https://bmcpregnancychildbirth.biomedcentral.com/articles/supplements/volume-17-supplement-2.

### Authors' contributions

GS conceived of the study, analyzed the data, and wrote the manuscript.

### Author's information

Department of Obstetrics, Gynecology & Reproductive Sciences, University of California San Francisco 1330 Broadway Suite 1100 Oakland, CA 94612.

### Ethics approval and consent to participate

This research is exempt. The UCLA Office of Human Research Protection Program determined that the research does not meet the definition of human subject research because the data are available through the Economic Research Forum and are de-identified.

### Consent for publication

Not applicable.

### Competing interests

The author declares no competing interests.

##

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
