## [Open peer review. (PDF 224 kb) · BMC Pregnancy and Childbirth]

Reviewer reports

Title: First birth and the trajectory of women's empowerment in Egypt

Reviewer 1 : Sarah Jane Holcombe

I enjoyed reading this paper, and found the topic and research questions to be important and well-defined. The methods were generally very well-explained, although more should be done to situate the design and new contributions of the measures of empowerment used. The longitudinal analysis is a particular strength of this research.

Major Compulsory Revisions

1. The title should be changed to reference the country (Egypt).
2. The author should do more to situate the contribution of her research with regard to existing research on the topic area, particularly that focusing on Egypt and the Middle East. Relatedly, there are earlier and concurrent studies of women's empowerment in Egypt (Kishor, 1995; Khatab Sakr, 2009; Abdel, Moula 2000; Assad, Nazier, Ramadan 2015), many of which also use the ELMS and use measures of mobility and decision-making to assess empowerment. While some of these are from the gray literature, they are nonetheless important.
3. Dependent variables – The author is very clear in describing the construction of the four measures of empowerment. However, she should provide the following types of clarification/justification:
 - A. Please justify why the outcome measure is empowerment in the second time period, rather than the difference between empowerment in the first and second time periods.
 - B. The author should characterize more clearly the significance of/rationale for the individual decision-making versus the joint-decision-making empowerment measures and how they are to be interpreted. Are they zero sum? Do they have an inverse relationship? How are they an improvement on previous such measures (e.g., Assad et. al. etc.)?
4. The author should explain why she uses 1 birth rather than 0 births as the reference category for births.

In the results/discussion,

5. In table 2, while the size of the increase in individual decision-making (from 2.48 to 2.54 decisions) is statistically significant, it may well not be substantively significant. The author should justify.
6. In table 3, the author should indicate whether there are statistically significant differences in the empowerment measures by parity.
7. In table 3, the author seems to have buried an interesting finding, that in these bivariate analyses, after 2 births, women's empowerment stays the same (financial autonomy) or declines (decision-making, mobility) – although we cannot tell if these differences are statistically significant. The author should note/explain what she makes, if anything, of the bivariate findings that empowerment peaks at 2 children, and then declines

(especially in the discussion section in relation to the multivariate findings). This is all the more notable as it applies to almost half the sample. Parity and its relation to empowerment plays out differently in the real world experience of women (bivariate findings in Table 2), and the hypothetical empowerment experience when other variables are held constant (multivariate findings in Table 4). The author should highlight this further in her discussion.

8. As the four measures of empowerment levels in 2006 are used as independent variables in the multivariate analysis, they should be presented (along with parity levels) in table 3 or in a new table. Specifically, a table should show the levels of empowerment in 2006 and 2012 among women going from 0 to 1 child, from 1 to 2 children, from 2 to 3 children, etc., between the two time periods.
9. The author found no significant relationship between parity and financial autonomy, and should discuss this, especially in light of the fact that the increase in financial autonomy observed between 2006 and 2012 (Table 2) appears to be the biggest of any increase in the empowerment measures.

	Change in measures of empowerment			Percentage increase
	2006	2012	Increase	
individual decision-making	2.48	2.54	0.06	2%
joint decision-making	3.57	3.05	-0.52	-15%
mobility	2.05	2.55	0.5	24%
financial autonomy	0.2	0.6	0.4	200%

10. In the discussion, the author should how her findings accord or contrast with those of other studies. For example, can she articulate how her findings fit into the study of the relative impacts of women's individual traits versus those of the overall social context (e.g., Mason and Smith), or what we should make of the finding that financial autonomy in 2006 was not a predictor of any of her empowerment outcomes (especially in relation to weight given to this measure in other studies)?

Minor Essential Revisions

The author might note that many have a working assumption that higher fertility means less empowerment for women.

11. Given that negative binomial regression is less intuitively obvious than OLS or logistic regression, the author should explain why it was selected (e.g., negative binomial regression because of an over-dispersed count outcome, etc.?) and how to interpret its results.
12. The analysis includes a measure of whether a woman was related to her husband. Could the Methods section spell out the implications of a woman being related to her husband? If the measure is intended to reveal empowerment, could the author explain why she did not use the age differential between the husband and wife?
13. line 4, 'gender equality' is not a strategy.

14. Please make this sentence more clear: “If empowerment is dynamic and changes over time based on exposures at the individual, household, and community level, it stands to reason that fertility would change later in life empowerment.”
15. The last item listed in the following sentence is not consistent with the two that precede it: “Women’s sexuality is used to restrict their mobility, access to financial resources, and other behavioral controls [12]. “
16. This is true in most societies – “...application to societies, like Egypt, where women’s social relations are integral to their identities [14, 15].”
17. Line 280 refers to table 5 – I believe this should be table 4.
18. Line 238: ‘fewer’ should be used rather than ‘less’.
19. Perhaps replace reproductive ‘decisions’ with ‘events’? It may not be the case that women perceive bearing their first child as their own decision or as a decision at all. “Reproductive decisions trigger a set of future decisions about employment, practices surrounding motherhood and childcare, and dynamics within a household.”
20. In the limitations section, the author should include the standard disclaimer that results cannot be interpreted as causal because of the possibility of effects from unobserved variables and that measurement bias may have occurred.

Discretionary Revisions

21. The article should reference the large and mature economics and demography literatures in the US that examine the relationships between women’s employment and fertility.
22. The background section could be strengthened by including data on trends in areas (education, income, age at marriage, etc.) that are included as control variables to mitigate confounding.
23. The author noted that the period of study was one of economic recession. Can she speak to whether she thinks her findings would hold in a period of economic expansion when their potentially might be more economic work opportunities outside the home available to men and women?
24. Could the author comment on the potential applicability of her measures of empowerment to settings outside of Egypt or the Middle East?

I declare that I have no competing interests.

Reviewer 2: Andrzej Kulczycki

The authors use a longitudinal design based on 2 waves of ELMPS data for 4,660 married women aged 15-49, along with a battery of female empowerment measures across several dimensions, and their associations with first and subsequent births over part of the life course. The paper makes for a potentially useful addition to the literature. The research questions are important, the methods are appropriate, and the paper is generally well written. There are no clear statistical flaws or errors in interpretation. Before proceeding with publication, however, I would recommend some revisions to the paper as follows:

Major compulsory revisions

This is generally a solid paper and the following major revisions are ones that should be done. These do not necessarily warrant changes to the analysis, but at least should be inserted in the text, especially the discussion, to strengthen the work:

- 1). the authors over-emphasize the study's longitudinal nature. The authors consider 2 panel survey waves over a 6-year period, which is valuable, but we do not learn more about how female empowerment evolves further over the life course and how it has changed across generations, if at all. These limitations should be acknowledged and it may be better to refer to data from 2 waves of the panel survey and that it is a 6-year panel investigation.
- 2). Regarding the Methods/Data section (lines 100-121), the authors should describe further the 2 panels, e.g. how many people were interviewed in each year, what proportion were married women in their childbearing years in 2006, what proportions of the total number of women interviewed had complete data on fertility and empowerment in 2006 and 2012, and how much attrition was there between the 2 survey waves. More importantly, were there any systematic differences between losses to follow-up and those who were interviewed in both panels?
- 3). The authors should also consider that some of the relationships may be reciprocal, e.g. childbearing may be a source of empowerment, but may also be a function of being more empowered, especially for more affluent/better connected women who already have 1 or 2 children. There may also be ideational factors, e.g. a desire to conform to social norms connected to women's gendered roles could be the driving motivation for having kids more than seeking greater empowerment. These factors are not considered in the analysis and should be mentioned as a potential limitation.
- 4). Some discussion of the association between fertility and women's empowerment in the Arab region would be welcome.
- 5). I would encourage the authors to consider age further, and in particular to consider how the results may vary for women who marry and have children at a young age, e.g. for married adolescents, compared to those who marry and/or have children at older ages, particularly after the median age at marriage. The authors have a large number of cases, so it would be useful to break down the age variable further.
- 5). I believe the ELMPS also included a survey round in 1998. Why was this not considered?

Minor Essential Revisions:

- 1). The discussions of empowerment in the Background and Measures sections are generally good, but it should also be mentioned that other dimensions of empowerment are not being considered. These include issues close to actual fertility outcomes, or proximate variables to fertility that are also gender-dependent, such as who makes decisions regarding contraceptive use and if the woman is able to refuse sex. This should also be stated as a limitation.
- 2). I would like to learn more about age at marriage and how these measures may vary by marriage timing, e.g. is there variation between women who marry at ages 16-20 compared to women who marry later? Further analyses could be conducted and would strengthen the quality of the work.
- 3). The authors should also discuss what other studies have shown that use the female empowerment measures in ELMPS-based studies related to fertility and those concerning topics other than just fertility.

I declare that I have no competing interests.

Response to reviewers

I thank the reviewers for their careful and thoughtful suggestions. I believe the revised manuscript is significantly improved as a result of their advice. This memo summarizes how I responded to the reviewers' comments. Segments of the reviewers' comments are reproduced in bold text, and my response to the reviewers' comments are inserted beneath in normal text.

Supplement Managing Editor: Ushma Upadhyay

As you will see, both reviewers wanted more on the gender context in which the study takes place. Further they wanted to understand the significance of your findings in this context. In your revision, please be sure to address these comments, contextualizing your study and its findings in the introduction and discussion sections.

Thank you for the positive assessment and additional suggestions for revision. The manuscript has been revised to contextualize the study further in the gender and overall context of Egypt in both the introduction and discussion sections. Several paragraphs were added to describe the household gender context of Egypt in the background. In the discussion, the findings have also been further situated in the community context of Egypt and the existing empowerment and fertility literature. Below, I respond to each of the reviewer comments.

Reviewer 1: Sarah Jane Holcombe

I enjoyed reading this paper, and found the topic and research questions to be important and well-defined. The methods were generally very well-explained, although more should be done to situate the design and new contributions of the measures of empowerment used. The longitudinal analysis is a particular strength of this research.

I appreciate the reviewer's positive assessment and suggestions for the paper.

Major Compulsory Revisions (The author must respond to these before a decision on publication can be reached. For example, additional necessary experiments or controls, statistical mistakes, errors in interpretation.)

25. The title should be changed to reference the country (Egypt).

The title has been revised to "First Birth and the Trajectory of Women's Empowerment in Egypt".

26. The author should do more to situate the contribution of her research with regard to existing research on the topic area, particularly that focusing on Egypt and the Middle East. Relatedly, there are earlier and concurrent studies of women's empowerment in Egypt (Kishor, 1995; Khatab Sakr, 2009; Abdel, Moula 2000; Assad, Nazier, Ramadan 2015), many of which also use the ELMS and use measures of mobility and decision-making to assess empowerment. While some of these are from the gray literature, they are nonetheless important.

Thank you for the additional suggestions for literature on the topic area. These citations from the gray literature as well as others were added to further situate the study in the Egyptian context and the Middle East and in the existing literature on women's empowerment more broadly (section on The Egyptian Context and the Discussion).

27. Dependent variables – The author is very clear in describing the construction of the four measures of empowerment. However, she should provide the following types of clarification/justification:

A. Please justify why the outcome measure is empowerment in the second time period, rather than the difference between empowerment in the first and second time periods.

This is an important point and one I have grappled with throughout the study. Given the methodological similarity in using both approaches and that both approaches produce the same results, after consulting with a statistician, I chose to predict empowerment at a second time period while controlling for the first. My reasoning is simply that a) I am interested in how empowered women are in 2012, b) logically it seems easier to interpret a predication of empowerment at time two, c) I believe it more clearly demonstrates how births at the earlier time period are associated with empowerment at a later time period, and d) it allows for an easier discussion of how a gain in one measure of empowerment in 2006 can be associated with a different measure of empowerment in 2012. Also, this method accounts for the change, demonstrates the direction of change, and acknowledges both the starting point of empowerment

in 2006 and the ending point in 2012. The manuscript was revised to include a sentence that sensitivity analysis of models of changes in empowerment between 2006 and 2012 produced the same results (lines 400 – 401).

B. The author should characterize more clearly the significance of/rationale for the individual decision-making versus the joint-decision-making empowerment measures and how they are to be interpreted. Are they zero sum? Do they have an inverse relationship? How are they an improvement on previous such measures (e.g., Assad et. al. etc.)?

The manuscript has been revised to include further explanation of the two count variables. The response categories of respondent, husband, respondent and husband jointly, in-laws, respondent, husband, and in-laws all together do not form a clear interval. Therefore, the two count variables capture the different ways in which women may make household decisions – alone or jointly with somebody else in the household. I believe these measures are an improvement over giving a greater weight to the value of a decision made alone vs. a decision made with a partner vs. a decision made with a group as in Assaad et al. 2015. When more value is given to women making decisions alone, there is an inherent assumption that more agency means making a decision on your own when this may not be the case. More agency may include the ability to discuss and negotiate decisions with a partner or family member. Therefore, the study includes both individual and joint decision-making.

The reviewer is correct that the measures are zero sum – if women do not make decisions alone or do not make decisions with somebody else, they make no household decisions. This is now stated in the description of the measures (lines 325 – 328). For sensitivity analysis, a combined count measure of all possible ways women make household decisions was created, which ranges from 0 to 10 and captures all possible household decisions women make. The multivariate results using the combined measure are very similar to joint decision-making results. The table is included as an appendix and is presented below:

Appendix Table 1. Negative Binomial Multilevel Regression Models of Combined Decision-Making in 2012, Egyptian Labor Market Panel Survey (N=4,660)

Key Variables	Combined Decision-Making 2012	
	Negative Binomial	
	IRR	(SE)
Births 2006 (Ref=1 Birth)		
0	0.85***	(0.04)
2	1.24***	(0.04)
3	1.42***	(0.06)
4+	1.53***	(0.07)
Combined Household Decision Making 2006	1.03***	(0.00)
Mobility 2006	1.10***	(0.02)
Financial Autonomy 2006	1.06*	(0.03)
Age (years)	0.94***	(0.00)
Years of Education	1.02***	(0.00)
Less than 18 years old at marriage	0.87***	(0.03)
Dowry (Ref=None)		
No Response	0.97	(0.03)
Some	0.99	(0.03)
Related to Husband	0.99	(0.02)
Ever Worked	1.09**	(0.03)

Region (Ref=Greater Cairo)		
Alexandria & Suez Canal	0.96	(0.06)
Urban Lower	0.96	(0.05)
Urban Upper	0.86**	(0.04)
Rural Lower	0.96	(0.05)
Rural Upper	0.77***	(0.04)
Household Wealth Index (Ref=Poorest)		
Poorer	0.94	(0.03)
Middle	0.93*	(0.03)
Richer	0.95	(0.04)
Richest	0.89*	(0.04)
Husband's Age (years)	1.00	(0.00)
Husband's Years of Education	1.00	(0.00)
Variance at Level 1 (Individual Level)		
Variance at Level 2 (PSU Level)	0.55	(0.09)

Notes: *p<0.05, ** p<0.01, *** p<0.001. Standard errors in parentheses

28. The author should explain why she uses 1 birth rather than 0 births as the reference category for births.

This is a good point to clarify. In order to avoid using the smallest sample category (0 births, N=458) and still show whether or not there is a difference between 0 and 1 birth, a reference category of 1 birth is used. In addition, as per the editors' request, using the first birth as the reference category also provides the opportunity to show whether there is a difference between a woman's first birth and any subsequent births. The description of the fertility measure now includes this information (lines 351 – 353).

In the results/discussion,

29. In table 2, while the size of the increase in individual decision-making (from 2.48 to 2.54 decisions) is statistically significant, it may well not be substantively significant. The author should justify.

In the discussion of results for Table 2, it is noted that the average empowerment for married women in 2006 and 2012 is similar. The revised manuscript includes an additional sentence that states while there may be a significant change between 2006 and 2012 for individual decision-making, women are still making an average of two to three household decisions on their own (lines 421 – 423).

30. In table 3, the author should indicate whether there are statistically significant differences in the empowerment measures by parity.

Table 3 has been revised to include the statistically significant differences in empowerment measures by parity. Given the revision of the table to include measures of empowerment in both 2006 and 2012 (see response to comment #8 below), the statistical significance was noted on the variable names as to not excessively clutter the table and make it difficult to understand both the differences by parity and the significant differences between 2006 and 2012. Although the relationship between empowerment in 2006 and parity in 2006 is not part of the primary research question, since Table 3 now includes empowerment in both 2006 and 2012, the statistically significant differences in empowerment in 2006 by parity were included for consistency.

31. In table 3, the author seems to have buried an interesting finding, that in these bivariate analyses, after 2 births, women’s empowerment stays the same (financial autonomy) or declines (decision-making, mobility) – although we cannot tell if these differences are statistically significant. The author should note/explain what she makes, if anything, of the bivariate findings that empowerment peaks at 2 children, and then declines (especially in the discussion section in relation to the multivariate findings). This is all the more notable as it applies to almost half the sample. Parity and its relation to empowerment plays out differently in the real world experience of women (bivariate findings in Table 2), and the hypothetical empowerment experience when other variables are held constant (multivariate findings in Table 4). The author should highlight this further in her discussion.

I agree with the reviewer that this is an interesting finding. The results section for Table 3 (lines 434 – 443) describes the findings that in 2012, the average number of individual decisions and mobility increase for each birth from zero to two and then, decline for three and four or more births. I also explore this finding in the discussion section (lines 558 – 570).

32. As the four measures of empowerment levels in 2006 are used as independent variables in the multivariate analysis, they should be presented (along with parity levels) in table 3 or in a new table. Specifically, a table should show the levels of empowerment in 2006 and 2012 among women going from 0 to 1 child, from 1 to 2 children, from 2 to 3 children, etc., between the two time periods.

As suggested, Table 3 was revised to include the measures of empowerment in both 2006 and 2012. A note also indicates that all changes in empowerment from 2006 to 2012 for women at each parity are significant ($p < 0.001$).

33. The author found no significant relationship between parity and financial autonomy, and should discuss this, especially in light of the fact that the increase in financial autonomy observed between 2006 and 2012 (Table 2) appears to be the biggest of any increase in the empowerment measures.

	Change in measures of empowerment			Percentage increase
	2006	2012	increase	
individual decision-making	2.48	2.54	0.06	2%
joint decision-making	3.57	3.05	-0.52	-15%
mobility	2.05	2.55	0.5	24%
financial autonomy	0.2	0.6	0.4	200%

This is an important point, and I include a discussion of the lack of a significant relationship between parity and financial autonomy in the discussion section. Given the large increase in financial autonomy observed between 2006 and 2012, I also rechecked the analysis and found an error in transposing the 2006 values. It is in fact much closer to financial autonomy in 2012. In 2006, 65% of women are financially autonomous, and in 2012 60% of women are financially autonomous. All the bivariate and multivariate analyses were also rerun to ensure no additional errors, and in fact, all results remain the same. As such, the results and discussion explore why financial autonomy in 2012 is not associated with parity, which seems largely due to the measure

of employment capturing some aspects of the relationship between parity and financial autonomy.

34. In the discussion, the author should show how her findings accord or contrast with those of other studies. For example, can she articulate how her findings fit into the study of the relative impacts of women's individual traits versus those of the overall social context (e.g., Mason and Smith), or what we should make of the finding that financial autonomy in 2006 was not a predictor of any of her empowerment outcomes (especially in relation to weight given to this measure in other studies)?

To address this important suggestion, the discussion has been revised to contextualize the findings relative to existing research. A section on community variation has been added to include a discussion of studies of individual attributes vs. the social context (lines 604 – 659). Financial autonomy in 2006 is a predictor of 2012 individual decision-making and financial autonomy in 2012. The revised manuscript reflects this finding. The discussion section also explores the financial autonomy results further (lines 576 – 589).

Minor Essential Revisions (The author can be trusted to make these. For example, missing labels on figures, the wrong use of a term, spelling mistakes.)

35. The author might note that many have a working assumption that higher fertility means less empowerment for women.

The revised manuscript includes a sentence on the underlying assumption that higher fertility may be associated with less empowerment over time (lines 71 – 72).

36. Given that negative binomial regression is less intuitively obvious than OLS or logistic regression, the author should explain why it was selected (e.g., negative binomial regression because of an over-dispersed count outcome, etc.?) and how to interpret its results.

To address this suggestion, the revised manuscript states that the negative binomial models are favored because of the over-dispersion of the count outcomes (lines 394 – 396), and that the incident rate ratios should be interpreted as a percent change in the rate (line 447).

37. The analysis includes a measure of whether a woman was related to her husband. Could the Methods section spell out the implications of a woman being related to her husband? If the measure is intended to reveal empowerment, could the author explain why she did not use the age differential between the husband and wife?

The age difference between the woman and her husband is also included in the analysis by including both a measure of the woman's age and her husband's age. The revised manuscript includes a discussion of the potential implications of endogamy for women's empowerment in the background section on Egypt (lines 122 – 126).

38. line 4, 'gender equality' is not a strategy.

The sentence was edited to include “promotion of gender equality” as the strategy.

- 39. Please make this sentence more clear: “If empowerment is dynamic and changes over time based on exposures at the individual, household, and community level, it stands to reason that fertility would change later in life empowerment.”**

The sentence was revised for clarity and now reads “Empowerment is dynamic and changes over time based on events and experiences at the individual, household, and community level; therefore, it makes sense that reproductive events, like fertility, would change women’s empowerment over the life course”.

- 40. The last item listed in the following sentence is not consistent with the two that precede it: “Women’s sexuality is used to restrict their mobility, access to financial resources, and other behavioral controls [12].”**

The sentence was edited for clarity and consistency.

- 41. This is true in most societies – “...application to societies, like Egypt, where women’s social relations are integral to their identities [14, 15].”**

The reviewer is correct. That portion of the sentence was removed and the sentence was revised to note the fact that measure of agency have been validated in the Egyptian context (lines 34 – 35).

- 42. Line 280 refers to table 5 – I believe this should be table 4.**

The sentence was edited to refer to Table 4.

- 43. Line 238: ‘fewer’ should be used rather than ‘less’.**

The sentence was edited to use ‘fewer’.

- 44. Perhaps replace reproductive ‘decisions’ with ‘events’? It may not be the case that women perceive bearing their first child as their own decision or as a decision at all. “Reproductive decisions trigger a set of future decisions about employment, practices surrounding motherhood and childcare, and dynamics within a household.”**

This is an important point, and ‘decisions’ was revised as suggested to ‘events’. This also aligned better with the use of ‘reproductive events’ in other places in the manuscript.

- 45. In the limitations section, the author should include the standard disclaimer that results cannot be interpreted as causal because of the possibility of effects from unobserved variables and that measurement bias may have occurred.**

The limitations section has been fully revised to include disclaimers on measurement bias and cautions against interpreting the results as causal as they are not (lines 671 – 705).

Discretionary Revisions (These are recommendations for improvement that the author can choose to ignore. For example clarifications, data that would be useful but not essential.)

- 46. The article should reference the large and mature economics and demography literatures in the US that examine the relationships between women's employment and fertility.**

This is an important literature for work on women's fertility. However, given the differences between the U.S. and the Middle East and North Africa and efforts to further situate this study in the Egyptian context, I believe the economics and demography literature on employment and fertility is outside the scope of this paper.

- 47. The background section could be strengthened by including data on trends in areas (education, income, age at marriage, etc.) that are included as control variables to mitigate confounding.**

The background section on the Egyptian context has been revised to include more data on employment and age at marriage in Egypt to explain the inclusion of some of the control variables (lines 90 – 158).

- 48. The author noted that the period of study was one of economic recession. Can she speak to whether she thinks her findings would hold in a period of economic expansion when their potentially might be more economic work opportunities outside the home available to men and women?**

While additional research would be needed to confirm whether or not the relationship between fertility and empowerment would remain the same in periods of economic expansion, the revised manuscript briefly explores this idea in the discussion section (lines 562 – 567).

- 49. Could the author comment on the potential applicability of her measures of empowerment to settings outside of Egypt or the Middle East?**

I believe the findings are only generalizable to Egypt. That said, I think the empowerment measures can be validated and used in other contexts.

Reviewer 2: Andrzej Kulczycki

The authors use a longitudinal design based on 2 waves of ELMPS data for 4,660 married women aged 15-49, along with a battery of female empowerment measures across several dimensions, and their associations with first and subsequent births over part of the life course. The paper makes for a potentially useful addition to the literature. The research questions are important, the methods are appropriate, and the paper is generally well written. There are no clear statistical flaws or errors in interpretation. Before proceeding with publication, however, I would recommend some revisions to the paper as follows:

I appreciate the reviewer's positive assessment and suggestions for the paper.

Major compulsory revisions

This is generally a solid paper and the following major revisions are ones that should be done. These do not necessarily warrant changes to the analysis, but at least should be inserted in the text, especially the discussion, to strengthen the work:

1) The authors over-emphasize the study's longitudinal nature. The authors consider 2 panel survey waves over a 6-year period, which is valuable, but we do not learn more about how female empowerment evolves further over the life course and how it has changed across generations, if at all. These limitations should be acknowledged and it may be better to refer to data from 2 waves of the panel survey and that it is a 6-year panel investigation.

This is an important point. Several instances of use of the word 'longitudinal' have been revised throughout to refer to data from two waves of a panel survey, and the discussion of limitations includes an acknowledgement of the limited nature of only using two waves that are six years apart. The research questions have also been revised to refer to 'over time' instead of 'over the life course' as the intention is not to mislead the reader that a full life course understanding of empowerment is possible with this analysis.

2). Regarding the Methods/Data section (lines 100-121), the authors should describe further the 2 panels, e.g. how many people were interviewed in each year, what proportion were married women in their childbearing years in 2006, what proportions of the total number of women interviewed had complete data on fertility and empowerment in 2006 and 2012, and how much attrition was there between the 2 survey waves. More importantly, were there any systematic differences between losses to follow-up and those who were interviewed in both panels?

The methods section has been revised to describe the sample selection in greater detail. The manuscript now includes the following information, "Of the 37,140 individuals interviewed in the 2006 ELMPS, 49% or 18,555 are women, 26.8% or 9,937 are in their childbearing years (between the ages of 15 and 49), and 6,296 women are married. Of the married women, 108 are missing data on empowerment and fertility in 2006 and 455 women are missing data on their spouses. Between 2006 and 2012, 1,181 women were lost to follow up, and these women are slightly younger, higher educated women from the greater Cairo area" (lines 232 – 296). I also include the following table of demographic characteristics for the married women who were lost

to follow up in the Appendix. The significant differences between the analytic sample and those lost to follow up are in age, education of the women and their spouses, and region. This is consistent with the analysis of attrition in the ELMPS (Assaad & Craft 2013) that showed young, educated people leaving the great Cairo region between 2006 and 2012. Despite this attrition, the 2006 and 2012 panel samples are comparable to the 2006 Egyptian census.

Assaad, R., & Krafft, C. (2013). The Egypt labor market panel survey: introducing the 2012 round. *IZA Journal of Labor & Development*, 2, 1-30.

Appendix Table 1. Fertility and Sample Characteristics (Means (SE) or %) of Lost to Follow Up Married Women Ages 15 to 49, Egyptian Labor Market Panel Survey

Key Variables in 2006	N	Married Women N=1,181 % or Mean (SD)
Number of Births		
0	201	17.02
1	324	27.43
2	288	24.39
3	191	16.17
4+	177	14.99
Mean (SD)	1,181	2.00 (1.69)
Gender of Firth Birth		
Boy	467	47.7
Girl	512	52.3
Current Age (years)	1,181	31.0 (8.29)
15 - 24 years	297	25.15
25 - 34 years	500	42.34
35 - 44 years	278	23.54
45 - 49 years	106	8.98
Years of Education	1,181	9.54 (5.54)
Age at Marriage (years)	1,181	22.2 (4.41)
Less than 18 years old	173	14.7
18 years or older	1,008	85.4
Value of Dowry		
No Response	291	24.64
No Amount	558	47.25
Some Amount	332	28.11
Related to Husband (1/0)	1,181	22.4
Ever Worked(1/0)	1,181	31.4
Region		
Greater Cairo	374	31.67
Alexandria & Suez Canal	217	18.37
Urban Lower	174	14.73
Urban Upper	132	11.18
Rural Lower	152	12.87
Rural Upper	132	11.18
Household Wealth Index		
Poorest	91	7.71
Poorer	178	15.07
Middle	236	19.98
Richer	330	27.94
Richest	346	29.3
Husband's Age in years	1,181	37.5 (9.93)
Husband's Years of Education	1,181	10.5 (5.32)

3). The authors should also consider that some of the relationships may be reciprocal, e.g. childbearing may be a source of empowerment, but may also be a function of being more empowered, especially for more affluent/better connected women who already have 1 or 2 children. There may also be ideational factors, e.g. a desire to conform to social norms connected to women's gendered roles could be the driving motivation for having kid more than seeking greater empowerment. These factors are not considered in the analysis and should be mentioned as a potential limitation.

This is an important point to make in the limitations, and the revised manuscript includes a discussion of the fact that empowerment prior to 2006 (which is not available in the data) could indeed be predicting childbearing, and that childbearing predicts empowerment in 2012 (lines 671 – 681). While having two waves of data begins to correct the temporal ordering issue in empowerment and fertility research, it far from solves the problem. Community norms around women gendered roles are likely a factor in childbearing; however, the data do not allow me to say whether or not this is a driving factor or how this factor compares to empowerment. This is explored in the discussion and addressed as a limitation (lines 620 – 637).

4). Some discussion of the association between fertility and women's empowerment in the Arab region would be welcome.

To address this suggestion, the background has been revised to include all existing studies on women's empowerment and fertility in Egypt as well as other studies of empowerment from Egypt (lines 127 – 158). The work in this area is very limited as most fertility and women's empowerment research focuses on south Asia and sub-Saharan Africa. However, the results were further contextualized for Egypt in the discussion and linked to any available information on women's empowerment and fertility in the region (throughout Discussion section).

5). I would encourage the authors to consider age further, and in particular to consider how the results may vary for women who marry and have children at a young age, e.g. for married adolescents, compared to those who marry and/or have children at older ages, particularly after the median age at marriage. The authors have a large number of cases, so it would be useful to break down the age variable further.

The revised manuscripts includes a further breakdown of age and age at marriage. In Table 1, the distribution of age in years and by categories of 15 – 24 years, 25 – 34 years, 35 – 44 years, and 45 – 49 years is shown. Table 1 also shows the distribution of age at marriage in years and by categories of less than 18 years old and 18 years or older. A quarter of the sample was married before the age of 18. The multivariate analysis includes the categorical variable for age at marriage in order to see whether or not there is a difference between women who get married before the age of 18 and those who get married after the age of 18. Results indicate that women who are married before the age of 18 by 2006 are less empowered in 2012 compared to those who were married at 18 years or older. At the same time, women who are older from 2006 to 2012 are less likely to make household decisions and have less mobility in 2012. The multivariate results indicate two separate trends that have to do with age: older women are associated with less empowerment over time, and women who married in adolescence (at younger ages) are associated with less empowerment over time.

5). I believe the ELMPS also included a survey round in 1998. Why was this not considered?

The reviewer is correct that the ELMPS was also conducted in 1998. However, the 1998 ELMPS does not include measures of women's empowerment or fertility and was therefore, not included in this analysis. The revised description of the data includes this information (lines 232 – 240).

Minor Essential Revisions:

1). The discussions of empowerment in the Background and Measures sections are generally good, but it should also be mentioned that other dimensions of empowerment are not being considered. These include issues close to actual fertility outcomes, or proximate variables to fertility that are also gender-dependent, such as who makes decisions regarding contraceptive use and if the woman is able to refuse sex. This should also be stated as a limitation.

This is an important limitation of the measures to address. The limitations section has been revised to include a statement about the lack of additional dimensions of empowerment that are more proximate to fertility (lines 684 – 685).

2). I would like to learn more about age at marriage and how these measures may vary by marriage timing, e.g. is there variation between women who marry at ages 16-20 compared to women who marry later? Further analyses could be conducted and would strengthen the quality of the work.

To address this important suggestion, the analysis has been revised to consider those who marry before the age of 18 and those who marry after the age of 18. There are significant differences in empowerment by age at marriage. Revised results are presented in Table 4.

3). The authors should also discuss what other studies have shown that use the female empowerment measures in ELMPS-based studies related to fertility and those concerning topics other than just fertility.

I thank the reviewer for this suggestion and the manuscript has been revised throughout to include additional work from Egypt using the ELMPS. The revised manuscript includes the few studies that have used the ELMPS empowerment measures (Assaad et al. 2015; Yount et al 2015) and others that have used the ELMPS fertility measures (Bertoli & Marchetta 2015).